# Being Nonreligious in Croatia: Managing Belonging and Non-Belonging

Nikolina Hazdovac Bajić

Institute of Social Sciences Ivo Pilar, 10000 Zagreb, Croatia; nikolina.hazdovac@pilar.hr

**Abstract:** Catholicism in the Croatian context has been one of the most powerful sources of collective belonging for centuries. Since the fall of socialism, desecularization tendencies have manifested as homogenization, collectivization, and deprivatization of religion. (Non)religiosity became a contested issue, which not only implied belonging (ethnic, national, historical) but was also highly politicized. This paper aims to explore how living in a society with a dominant collective religion influences the experience of nonreligious people. The conducted research was based on 30 semi-structured interviews with people who self-identify as nonreligious, but at the same time are not members of nonreligious organizations. The obtained data show that some elements of collectivism can push individuals away from religion, but for some nonreligious people, religiosity remains an important identification framework. Keeping a connection with religion is achieved through conformist behaviors or "cherry-picking" elements of religiosity, which are then combined in individually-consistent worldviews. Nonreligious people sometimes feel "left out" and experience their nonreligiosity as lonely and isolating, which they often do not want to pass on to others. This creates a specific position for some nonreligious individuals that is simultaneously "in" and "out" of religion, and challenges the way nonreligiosity is often imagined.

**Keywords:** nonreligiosity; collective religion; Catholicism; Croatia; conformism; religious belonging

## 1. Introduction

Scholars' interest in the phenomena of nonreligious and secular worldviews is increasing in parallel with the increase in the number of nones, nonreligious, and secular individuals in Europe and North America (Baker and Smith 2015; Furseth 2018; Zuckerman 2009). Most of the scientific production on these phenomena revolves around organized nonreligious, secular, or atheistic communities; collective or individual identity formations, and identity management in the light of various social prejudices against nonreligious people and atheists.

Since the very attempt to classify religion and it's other counterintuitive, it became clear that, in most cases, imposed classifications obscure the richness of substantive and affirmative forms of nonreligiosity, as a living reality that is performed in everyday life, in communities of practice, in forming individual identities and shaping interactions, in structuring social and political relations, and so forth. In 1972, Campbell (1972) described the religious–nonreligious continuum with a series of degrees of (non)religiosity, and since then, various scholars have sought to explore this "murky middle", pointing out that it is not a dichotomy with quantitative differences, but rather a field which comprises many layered and plural ways in which religiosity and nonreligiosity are combined, experienced, and manifested in everyday life (Lee 2015; Quack 2014). On the other hand, nonreligiosity is not only dynamic but also reactional and relational with respect to religion (Cimino and Smith 2014; Hout and Fischer 2002; Taylor 2007). Hence, although there are some basic similarities across societies, nonreligiosity is in many ways context specific.

Croatian society is highly religious. The different research works conducted since the 1990s, despite different methodologies, samples, and instruments, show a high level of reli-

giosity in Croatia, according to all indicators (Baloban and Črpić 2000; Baloban et al. 2014; Črpić and Zrinščak 2005, 2010, 2014; Nikodem and Zrinščak 2012, 2019). The Pew Research Center positions Croatia as the seventh most religious out of 34 European countries.[1] Despite high levels of declared religiosity and religious belonging in Croatia, other processes such as individualization and (contextual or differentiated) secularization can also take place simultaneously.[2] Due to specific socio-historical circumstances, religious and national identities in Croatia are intertwined, while the dominant Catholic Church is very publicly present, with a strong influence on the media, education system, and public discourse. Jakelić (2010) identified this type of religion as collective and described it as culturally specific, historically embedded, and defined in opposition towards the religious Other. In the Croatian case, it is also defined partly in relation to the "internal Other" (Zubrzycki 2006, p. 54) or "inside-outsider" (Trzebiatowska 2021): someone who lives within a society but is never fully accepted because she does not fit into the collectivity on a national or religious basis. This position of "internal Other" was reinforced after the collapse of socialism and disintegration of Yugoslavia, when the dominant social pattern took a turn toward retraditionalization, recollectivization, and homogenization, in the national and religious sense. This paper aims to explore how living in a society with a dominant collective religion influences the experience of nonreligious people. Although non-religiosity is in such a context often self-conscious and charged (Cimino and Smith 2007, 2014; Hout and Fischer 2002; Stahl 2010), this is surely not its only form. In that sense, this paper seeks to answer the following research questions: (1) which elements of collective religion, if any, interfere or affect the process of leaving religion; (2) how does context with collective religion influence the behaviors, practices, and beliefs of nonreligious people; (3) how do nonreligious people manage their "internal Other" status or their perception of belonging and non-belonging.

## 2. Croatian (Non)Religious Landscape

Since the initial process of embracing Christianity in the period from the 7th to 11th century, the Catholic Church has played an important role in creating and maintaining the collective identity of the Croatian people. During the war with the Ottoman Empire in the 14th and 15th centuries, Roman popes referred to the Croatian territory as *Antemurale Christianitatis*; towers and walls that protect Christianity. The existence on three religious borders (Catholic, Orthodox, and Islamic), shaped specific Croatian experiences, in which religious identity often served as a guardian and protector of national values. During this turbulent history, Croatia was part of other empires and state formations, except in the period from 1941 to 1945, during which a short-lived quisling Independent State of Croatia (NDH) was established, under the Ustaša regime and supported by Axis forces: Hitler's Germany and fascist Italy. The Catholic Church at this time had a great role in the public life of the new state (Perica 2002, pp. 24–25) and welcomed it as the long-awaited realization of the Croatian national independence (Jakelić 2010, p. 116). The Church position in NDH is still to this day problematic for part of the Croatian public, especially since the official Church reluctantly mentions Ustaša's crimes and some members of the clergy relativize them.

During the socialist period, in a multi-religious and multi-national Yugoslavia (1945–1991), the official government of the time saw religion as a negative and backward social factor. Religion and Church were suppressed from public life, but, although an "ideological struggle" was fought against them (with various intensities over time), especially through the media and education system, they remained present in the traditional family patterns of behavior and socialization, celebrations of patron saints, holiday gatherings, implementation of local religious traditions, etc. (Jukić 1994, p. 366). In this way, religion was not publicly visible, but still served as a collective identification mark. Data show that over 40% of Croatian citizens declared as religious during socialism, while an even higher percentage of citizens declared a confessional affiliation (Marinović Jerolimov and

Hazdovac Bajić 2017, p. 116). This means that part of the nonreligious people also declared belonging to some (mostly Catholic) confession at the time.

After the collapse of socialism and the introduction of democratic changes, Croatia experienced military aggression. The modern Croatian state was, thus, created during the war, while its democratization and social consolidation were based on homogenization in the national and religious sense. During this time a strong connection between different national and religious identities in the region was present: Croat–Catholic, Serb–Orthodox, and Bosnian–Muslim. The newly established political elites based their legitimacy, as well as the legitimacy of the new social system, partly on Catholicism as a historical guardian of traditional and national values and as the only social factor that represented opposition during socialism. The relationship between the Catholic Church and the state was regulated by the constitution and four agreements with the Holy See. Part of the public claimed from the very beginning that these, so-called Vatican Contracts, put the Church in a privileged social position.[3]

(Non)religiosity in these circumstances became a contested issue, which not only implyied belonging (ethnic, national, historical) but that was also highly politicized. The Church entered the public arena, became a powerful social factor through its connection with the ruling right-wing party, it entered the education system on all levels (public schools with the introduction of confessional education, universities, and preschools), the media, family life, and sought to influence legislation by associating with a range of civil associations.[4] The dominant national narrative almost always includes religion. Public state holidays are celebrated with Holy Masses, and religious holidays include state leaders at the forefront of the churches. Some of the leading politicians since the nineties declared as atheists or agnostics, but not without some doubts in the public about whether someone who is a nonbeliever can lead a Catholic country.[5] Narratives connected with widely spread folk religiosity also include national elements, such as in the popular saying "God and Croats" or the well-known religious song "Heavenly Maiden, Queen of Croats". The Catholic Church in Croatia and its clergy often invoke the dangers of atheism, Godlessness, and the ever-present hovering communism. As religiosity became a dominant and conformist position in the society, there was a considerable increase in declared religiosity among Croatian citizens and a decrease in the number of those who declared as nonreligious or atheists (Aračić et al. 2003; Marinović Jerolimov 1993). The great majority of Croatian citizens (about 90%) report confessional adherence, while almost 80% of them declare themselves as religious (Nikodem and Zrinščak 2019).

While the religion of the traditional, public, deprivatized, and collectivistic type with strong national character remains an important symbolic identification framework for the majority of the Croatian population, there are, at the same time, more or less subtle indications of individualization and secularization. This can be traced through the high confessional identification and strong presence of elements of traditional religiosity (mainly in family socialization, public expressions of religiosity such as the celebration of religious holidays, attending religious education, and attaining sacraments), with other indicators of religiosity that are not as present. In other words, there is a graduation, from almost complete confessional identification in the society, toward a lower level of religious self-identification, followed by a lower level of acceptance of fundamental religious beliefs, and even lower level of religious participation, with the acceptance of moral Church doctrine at the end (Črpić and Zrinščak 2014). Furthermore, there is a visible trend of losing trust in the Church, with an increasing number of citizens who think that the Catholic Church in Croatia is a rich institution, primarily interested in power, and with little capacity to help in solving moral or spiritual problems (Nikodem 2011; Nikodem and Zrinščak 2019). Data show that about half of the religious population in Croatia follows the Church's teaching, and the other half expresses more individualized forms of religiosity (Marinović Jerolimov 2005; Nikodem 2011). All of this causes reactions from the clergy. In one public holiday sermon before the Prime Minister and the President, one of the Croatian bishops complained that the practical and theoretical atheism, religious

indifference, and avoidance of Church belonging were an elite phenomenon in the previous system, while, in this system, religious illiteracy of young people and secularism entered Croatian homes.[6] At the same time, nonreligious organizations and civil initiatives started to emerge in the Croatian public sphere. Although they gather a small number of people, they have the potential to attract a larger number of sympathizers, and even religious people, regarding specific debated themes, which are in the focus of cultural wars (abortion, immigrants, health, and religious education in public schools, etc.).

Hence, the religious changes in Croatia could be described using religious complexity as a theoretical concept, which refers to the coexistence of several religious trends that take place at different levels of society, with different intensities and different (even opposing) directions (Furseth 2018).

### 3. Conceptual Background

Although nonreligiosity has gained considerable scholarly interest, there is still a lack of research, especially qualitative, in the societies outside Western Europe and North America. Studies in these countries have shown that the form of culturally relevant religion(s) is closely connected to the forms of nonreligiosity and the identities of the people who reject it (Smith 2011). In other words, nonreligiosity is dependent on perception or experience related to religion. Furthermore, they showed how nonreligiosity is a highly complex and polysemic phenomenon at the individual, group, and institutional levels (Beaman and Tomlins 2015; LeDrew 2015; Cotter 2011; Lee 2015; Silver et al. 2014). A great deal of the existing studies focused on the organized forms of nonreligiosity or atheism, while the "quiet majority" of nonreligious people, with less coordinated and more diversified expressions of nonreligiosity, are still understudied. Differently put, there is still a lot to be learned about nonreligiosity and how it is "actually lived, expressed, or experienced" in everyday life (Zuckerman 2010, p. viii).

Religion in Croatia can be described as collective, and religious identity as ascribed and not chosen (Jakelić 2010). On the other hand, separation from religion or the Church is a conscious act for the majority of those who define themselves as nonreligious in Croatia. This separation is rarely sharp and clear. Existing studies from Western European and American societies suggest that the main reason cited for rejection of religion is its inconsistency: logical, value related, or moral (Zuckerman 2015). This is confirmed in the study conducted by Trzebiatowska (2021) in the Polish context, which shows striking similarities to that in Croatia.

Research among members of nonreligious and atheistic organizations conducted in Croatia (Hazdovac Bajić 2019) concluded, similarly, that interviewees' separation from religion was explained based on intellectual and value disagreements, as they accept a different set of "secularly sacred" values (Knott 2010). Members of these organizations expressed their feeling of stigmatization due to the normative position of religiosity in the society, the ubiquity of religion in the public space, and the politicization of issues of (non)religiosity. Since these individuals actively express their nonreligiosity and reconfirm it through involvement in these organizations, their break with religion is clearer and their attitude towards the Church and its public activities is almost entirely negative. Data gathered in the International Social Survey Program (ISSP) in 2018 indicate that there is a much bigger diversity among nonreligious people. Only 54% of the surveyed nonreligious people do not belong to any church or religious organization. Less than half of them declare as atheists, with about 20% as agnostics, about 15% believing in some higher power, and about 10% expressing uncertainty about God's existence, while about 5% believe in God. As such, nonreligious people in Croatia retain a certain connection with religious traditions and beliefs, but also with the Catholic Church itself, much like studies in other countries have shown (Cotter 2015; Day et al. 2013). Cotter (2011) argued that nonreligious people express various combinations of seemingly incompatible (non)religious self-representations, identities, practices, and beliefs, which are brought together and consistently interpreted on an individual, practical level. These interpretations are also fluid and contextually variable.

Based on the data that indicate that nonreligious Croatians maintain certain religious elements, which creates their position as "inside Other", this paper will try to demonstrate how collective religion does not only influence "nonreligiosity" by the virtue of what is being rejected, but also in providing different aspects which are chosen to be kept.

## 4. Methodology

This study focused on the "quiet majority" of nonreligious people, for whom non-religiosity is not necessarily pushed into the foreground of their identities or daily lives, but just exists as a repository that can be more or less important or brought to the fore depending on the situation. The conducted research was, thus, focused exclusively on the people who self-identify as nonreligious, but at the same time are not members of nonreligious organizations. The research was based on a qualitative methodology. I conducted 30 semi-structured interviews. This method allowed informants to discuss, reflect, and describe their experiences in a conversational style. Interviews were held face to face or were computer-mediated, since the research took place during the pandemic. Before the interview, I asked interlocutors for verbal informed consent, whereby I informed them about the subject, purpose, and objectives of the research, the ways of using the information collected, methods of protecting anonymity, and the voluntary nature of participation. I recorded all 30 interviews with the oral consent from interviewees, and later transcribed them manually and coded them using the software program NVivo. Interviews lasted about 60 min on average. During transcription, I translated all individual speech characteristics (such as dialect features) into standard Croatian, to protect the anonymity of the interlocutors. For the same reason, I changed the names and other information that might indicate identity. Audio records, transcripts, and all the other data were later stored in an archive to which only I have access.

The analysis process had several steps. After transcription, the material was encoded line by line. The initial codes were grouped into categories and later into themes (Braun and Clarke 2006). During this process, three distinct themes emerged: (1) distancing elements of collectivity (2) maintaining an ambiguous relationship with Catholicism or the Church, and (3) managing non-belonging and internalized "otherness".

In the sample selection, I used convenience and snowball sampling. On social media, I announced an invitation for participation in the study. At the same time, I announced an invitation on public forums about nonreligiosity and atheism, and used other contacts to advertise it in their informal groups. The final sample consisted of 17 male and 13 female interviewees, 18 of them had a university degree, 3 held doctorates, and 9 had finished high school.[7] Ages ranged from 20 to 74, with a mean of 43 and a median of 39. Out of the 30 interviewees, 20 claimed that they were raised in religion, and the remaining 10 that they were raised non-religiously, which means that during their childhood they never went to church or other religious institutions and they were not taught beliefs or practices of a religious tradition by their family members. Nevertheless, two nonreligiously raised interviewees were baptized as children, which amounts to 22 baptized people in the sample. Only one of them is considering starting the formal process of leaving the Church. Regarding their confessional identity, 9 people in the sample would identify themselves as Catholics. Although all interviewees declared as nonreligious, they used this term as a superordinate term that comprised a multitude of other identifications: 9 of them self-declared as atheists, 6 as agnostics, 5 as spiritual but not religious (SBNR), 1 as a rationalist, 4 of them claim that they are simply non-believers, and 5 express insecurity about labels, while describing fluid identities which comprise elements of atheism, agnosticism, and spirituality. All of the interviewees declared themselves as Croats, and 10 of them come from ethnically mixed marriages.

## 5. Leaving Collective Religion

Abandonment of collective religiosity does include "typical" pathways to nonreligiosity based on the intellectual, moral, and value illogicality already described in the

literature ([Knott 2010](); [Smith 2011](); [Zuckerman 2015]()). Some of the interviewees, thus, cite their rationality as the main reason for rejection of religion, such as Ana (43), who had a desire to be religious, but just "couldn't come to terms with it". She jokingly compared it to network marketing, because everyone around her tried to assure her "how great it would be" and how she "would achieve great benefits", but she concluded "although I would love to, my mind simply could not accept it". Other interviewees mention logical inconsistency in religious truths that have "nothing in common with the modern knowledge of mankind" (Fran, 20). The hypocrisy and moral inconsistency of the clergy and the Church are also described as important deal-breakers for some nonreligious people. Mira (47) claimed:

> "I wanted to [believe], I was not born to be against or the opposite. However, for me hypocrisy and the fact that they are saying one thing and doing the opposite ... I was fed up with that sentence that they're just people. Well, they are people, but people who have chosen to be honorable".

Another common cause of contention throughout the interviews was adhering to a different set of (secular) values that are brought into opposition to those expressed through religion. Neda (67) expressed her strong feelings about it:

> "Although I tried, I grew more and more disappointed in the Church and its views in the political sense, in all matters ... They almost always had the view which was the opposite of mine: abortion, sexual minority rights, attitude towards national minorities, toward women, nationalist exclusivity in Croatia, leaning towards a far-right policy. I can't stand it".

The cited quotations indicate a strong religious context that can put a certain degree of social pressure on the individual. The interviewees above used different disclaimers, such as that they "wanted to" believe, they "tried", they had "the desire" and were not "born against or opposite" to show that they have made some effort to "be religious".

Other interviewees described various other causes of disengagement from religion that shed light on negatively perceived aspects of collectivity. For some of them, religiosity was an external "coercion" to which they were exposed and did not want to submit. Ivo (39) claims "I had to go because everybody else went [to church]. But as soon as I could decide for myself, I stopped". Robert, who was raised in a very religious family explained:

> "My upbringing was fiercely religious and patriarchal. In fact, I was forced to go to Mass every Sunday. When I was younger I just imagined when I would be an adult that I no longer had to go. I was waiting ... I was overjoyed when I got to the stage that I no longer had to do all those things. To me that compulsion when I was younger ... I would say it completely pushed me away from religion and the Church". (Robert, 41)

For many other interviewees, besides Robert, reflecting on religion includes the use of specific terminology (coercion, compulsion, extremism, forcing, imposition, bothering, nagging, and so forth) that implies different forms of social imposition of religiosity that they openly or intimately reject. Lucija (37), who during her interview emphasized several times that she still believes in God, said that she was also put off by the compulsion: "I didn't see the point in that. In that institution, in those customs and rules that forced me ... that I had to do something ... " Some other interviewees described their rejection of religion based on its banalization in collective representations, in which it became nothing more than an empty signifier of belonging. Sonja (39) said:

> "I would say that society had a lot of influence on me, or rather my counter attitude to society. I think that if society hadn't been as religious as it was, maybe some things would have interested me more. I would say there was a lot of resistance in me against that trivialization of religion by society as one mass, as the herd".

Other interviewees described similar negative feelings toward the "empty Catholicism" (Neda, 67) of people who do not care about "true spirituality" but attend church "because it is a sign of the social prestige" (Petra, 31).

On the other hand, the "normalization of Croat-Catholic"[8] (Marija, 44) in a society with collectivistic religion implies that members of other ethnic groups are excluded. Some of my interviewees describe their experience of "otherness":

> "As a child from a mixed marriage, I was rejected and I didn't feel welcome. I am from a small community and everyone knew who I was, so I always had some discomfort around priests and religious teachers in school. They used to suggestively ask me something, like what was I doing there or since when do I celebrate something. But other people used to do that too, including school teachers. It was more of the nationality issue, but it always came to the fore or as problematic around the issue of Church and religion". (Zoran, 39)

Nikola (48) has a similar experience: "The division is so great that . . . I felt their intent to expel us, because we were not part of 'the clan', simple mistrust, even though they knew me my whole life".

At the same time normalization of religiosity and life in religiously homogenous communities minimizes the questioning of the usual pattern. Some interviewees recollect their first encounter with people from other religious traditions and without religion as an important element of their break with religion. Lucija (37) said: "There wasn't one person of different religion in my elementary school, we were all Catholic. When I came to high school, I had a few Muslims in my class. I think that changed my perspective". Similarly, Jelena (27) remembers starting university in a bigger town and meeting "other students and friends who were not believers" as an experience that changed her views.

Apart from the different paths toward nonreligiosity, described above, nonreligious people continue to be a part of a religious society. In the following sections, I try to describe how living in a society with a dominant collective religion influences the experience of nonreligious people, using two themes: maintaining the relationship with religion, and managing non-belonging or "otherness".

## 6. Maintaining an Ambiguous Relationship with Catholicism or the Church

Not only because of the religious environment, but also for various other reasons, religion can play an important role in the lives of nonreligious people. Being nonreligious does not confer an automatic set of beliefs or practices. Through interviews, I noticed various situations in which interlocutors decided either to "keep one foot in the door" of the religion or to "pick and choose" among elements of religiosity. In addition, their behaviors can differ over time and depend on the setting. For some interviewees, conformism was an often used tactic in social settings. Petra (31) was raised in a religious family and works in a highly religious environment. Although her partner is not religious, she still struggles to express her views publicly. She explained:

> "Even when I had already rejected religious beliefs, I attended Mass because it was simply conformism on my part. I work as a teacher and every now and then there is some event that requires us to attend the Mass. It is something that is expected from me. It isn't mandatory and I could refuse it, but I don't want to explain myself, I don't want that kind of attention".

Mirjana (53) says: "I sometimes attend Mass. It's my childhood upbringing. But that doesn't mean anything to me right now. I am completely empty in these situations". The impression that broadly accepted religion is just "quasi-Catholicism" (Jana, 54), where everybody has "a cross around the neck" as a "recognition mark" (Marin, 33) and not because they truly believe or accept the Church's teachings (especially in the field of moral issues), makes it easier to blend in and "fake" or conform in different situations. Mirjana (53) explained that she does not feel "out of place" while attending Mass because for most of the people around her religion is "empty, without content and just to show".

Another example of conformism is selectivity in disclosing one's nonreligiosity. The interviewees claimed that: "Realistically, sometimes it is better to keep it quiet" (Luka, 44) or "It is not a hill I would die on, so to say" (Robert, 41). Others, too, admit that (non)religiosity is not "the topic I discuss a lot. Yes, with some friends I may get into a deeper conversation, but these are people I know I can talk to" (Goran, 51). Mladen (40) openly says: "My first reflex is to say that I would be open about it, but there is something called 'read the room', especially if we are talking about a sensitive situation. I'm not above distorting the facts a little".

From all these examples, it is obvious that some interviewees, in certain social situations, keep a "religious" facade. (Non)religiosity is not openly discussed in society, or sometimes even among families: "My family is openly Catholic . . . I don't know . . . we don't talk about it often. I don't know if I would tell them openly" (Fran, 20). These examples show split identities: the external, which is used for keeping a symbolic connection with the wider community; and the internal, which is more in sync with an individual's outlook.

A specific situation occurs among nonreligious parents with younger children. Some of the interviewees simultaneously try to pass the collective religious identity to their children (even though they reject it) and the secular values they consider important. Marija (44) claims:

"Concerning the children I have a specific approach. I have already said, they go to religious education, they are baptized, and they have received all sacraments so far. Since both, my husband and I, are nonreligious, this may be an unusual situation. But, you know what? I didn't want to limit them in life with my nonreligiosity. It may be conformism on my part, but I didn't want to deny them the community and tradition".

She also continues:

"I think, however, that they will never accept everything that religion claims because I teach them that they should respect everybody, no matter of their sexual orientation, race, or nation. [ . . . ] I never tell them that they should fear God or do good out of fear of God's punishment, but out of their human responsibility. I also teach them that every person is the master of her body and many other things that are probably not in accordance with the Church's teachings, but I think are extremely important".

The effort to raise children within a symbolic religious belonging, to build an external impression that will allow them to decide on it later, reflects the fear of being different in a context that relies on collectivity: "And then you realize that it's best to fit into the community. I wouldn't prosecute justice on behalf of my child. It is much easier to fit in and to be less noticeable" (Lucija, 37).

Keeping elements of religiosity is not only a matter of conformism or keeping a religious facade. Some interviewees express opposite attitudes regarding faith and religious institutions: "I think that anyone who has read the Bible cannot say anything against the 10 Commandments of God and all those messages that Jesus sends. But people have turned it all upside down and ruined it" (Mirjana, 53). This rationalization through which religious beliefs are seen as a common good, which people corrupt, makes it possible to pick and choose between elements of different religious traditions and eclectically combine them with elements of spirituality. Some interviewees reveal that they hold onto different elements of religiosity, whether because they were so deeply engrained in them or because they are the closest outlet for spiritual seeking that they have. Jelena (27) sometimes finds "peace, feeling of holiness, meditation, and spirituality" during the Mass, Lili (63) finds "deep spiritual feeling" often during prayer and experiences relating to God as "relating to everything from grass to clouds", and Lucija (37) "definitely believes in God" but also in "some sort of reincarnation".

There is a deep appreciation for some elements of religion, which are seen as the cultural base of the society: "Western society and everything that makes up our morals, it all stems from Christianity. So our society has deep Christian roots" (Sonja, 39). Religion is not only appreciated as a "cultural foundation of our society" (Goran, 51), but also as a key factor of Croatian identity: "The fact is that a lot of Croats survived and felt safe during socialism, especially in the diaspora, by connecting to the church that other Croats went to" (Lili, 63).

Some interviewees, on the other hand, express awareness that religious concepts were so deeply ingrained in them through socialization that they cannot "get rid of them" (Ivo, 39). During hardships, few of them admit they cross themselves (Petra, 31) or "turn to God" (Zoran, 39). Petar (37) who, like some other interviewees, was raised in a nominally religious family, which was from the outside seen as close to the Church and sacramentally orientated, while most of its members were not intimately very religious, claims:

> "I am in the process, somewhere halfway between what was implemented in me at the earliest age . . . and I need active strength of mind to resist these, I would say, concepts that were installed in me—heaven, hell, what is good and what is bad, sexuality, non-sexuality, chastity, and all the other stupid things they instilled in me".

Petar thinks that he managed to change his beliefs to the extent that he now considers himself a nonreligious person, but continues:

> "That does not mean that I am a non-spiritual person. I think that religion as a concept has some institutional and political connotations, identity as well, especially given the nationality I belong to, Croatian. Concepts that were installed to me do not meet my needs, so I consider myself a nonreligious person and have absolutely no tendency to seek spiritual meaning within some institutions but exclusively within myself. But that doesn't mean that I'm immune to nationality and national belonging".

Mladen (40), although declaring as agnostic and rejecting religious beliefs, ties his identity firmly to Catholicism: "For me, Catholicism is an important part of my identity, it is a culture and tradition that I grew up in". For some other interviewees the concepts of nationality, religion, tradition, and culture are intertwined. Igor (30) describes his ambivalent feelings:

> "I have this problem that I love the cultural part of the religion in which I grew up. I love that music, art, architecture, all the customs, sense of belonging, but often I don't like the messages I hear from the Church. This creates a conflict within me. To put it bluntly, I can't ignore that part of my identity, and yet it irritates me in all sorts of other intellectual matters".

These quotations show how some nonreligious people try to "cherry-pick"[9] among different elements of religion, but without completely distancing themselves from the religious framework as the main feature of collective belonging.

## 7. Managing Non-Belonging

There is a common assumption in Croatia that nonreligious people are either advocates of the past socialist system or not ethnic Croats. To counteract this prejudice, some interviewees, although they position themselves outside of the religious collective (at least intimately, if not publicly), tried to firmly declare in the political and national sense:

> "Religion is here one mass that has its gravity and its sucks in other particles. People who do not want to be grouped into this mass simply have to find another way. We [in Croatia] have had half a century of a completely wrong social order—socialism. I have to stress that I am not a Yugoslav, I am not a socialist".
> (Petar, 37)

At the same time, it was important for some interviewees to emphasize their former "insider" position. Ivo (39) claimed that, "because of a religious upbringing", he has a "better understanding of the Church"; Mirjana (53) reasons that children should not have religious education in schools, but only in church "where I also went"; and Marija (44) simply states that "she was part of it all". Stressing their former "insider" status provides a link to a religious collectivity (which can be activated depending on the situation) but also emphasizes that distancing from religion was a conscious act and one's own choice.

However, some interviewees expressed a pronounced feeling of non-belonging. This sentiment was especially strong among interviewees that came from mixed marriages, but was also present among others. Some of them internalized this feeling to the point that they express some dose of inferiority. Nikola (48), debating on whether he could have a religious partner, said: "If she was religious she couldn't have loved me". Others show a willingness to indulge their surrounding in religious matters. Tibor (34) explained that he sometimes "attends Church things" but always "for someone else, because sometimes it's important to other people, and I don't care". Sonja (39), who also comes from a mixed marriage, said that she received her first communion only when she was 18:

"My mother turned more to religion and wanted to marry my father in the church. She, of course, had to promise to raise her children in the faith. The whole thing was surreal, but I have done it for her. It meant a lot to her, and it was all the same for me".

Marin (33) also received his sacraments later in childhood "under some pressure from society, friends, school" because he "wanted to be more like them". For some interviewees adjustment to religious surroundings evokes negative feelings. Mislav (54) claims "I was married in church. I attended hundreds of those things. I have compromised my views out of sheer decency to respect someone to whom these rituals were more important than to myself".

The specific way of managing nonreligiosity in a religious setting is also evident among those interviewees who agreed to raise their children with religion for the sake of a religious family or partner, to whom it was important. Ana (43) explained her situation:

"As for the baptizing my children and religious education, even today, I must admit, I have a problem with that. I don't like it at all. But for the peace in the house, I agreed. My children know that I am nonreligious and I redirect every question related to their faith to their father".

Similarly, Zoran (39) said:

"My wife is from a religious family and she wanted the children to have the sacraments and go to religious education. I agreed with that, but I don't want to have anything to do with it. I have no problem with having children raised like that, but I am not participating in it".

Internalized nonbelonging is not just visible in close relationships. It is also apparent as the feeling of being left out of the community and society. Luka (44) claims he "wished to be able to experience that belonging" and Zoran (39) thinks that his nonreligiosity is "more a disadvantage than an advantage, a handicap in the society" in which he lives and has he "felt it many times". An individual's nonreligiosity is very noticeable in religious surroundings Lucija (37) says, and adds "a good thing that I'm not of some other faith". Sonja (39) openly admits:

"I think it would have been a lot easier for me when I was growing up to be able to belong. Because I haven't belonged. There was a period when I wanted to, and for me, that belonging to the local community was very important".

Nonbelonging is not just an emotionally difficult experience. Interviewees also mention social capital that they lost or did not experience through religious participation. Robert (43) stresses the importance of community belonging in Croatian society and sees the church as the intersection of various social contacts: "Priest in my church knew everybody and

could have taken care of everything. So, if you went to Mass, he could have arranged you the internship, the doctor's appointment, the job, whatever". Similarly, Dario (33) sees the church as a source of social connections:

> "People who are close to the Church, who often go to Mass, can make acquaintances with those who are in positions or who are powerful. For example, all the politicians are great believers and every Sunday they present themselves at Mass. So, I think it is desirable to be a believer, it is easier to make your way through society".

Through all these examples, the interviewees showed that religion, or rather lack of it, is an important social factor. In a collectively religious Croatian context, it is not (only) a matter of private beliefs and practices, but also a public recognition. Owing to that, it is not (only) a matter of the individual, but also includes family and social relationships that have to be taken into account. This is confirmed by the fact that they are not ready to initiate the very public process of formally leaving the Church.

## 8. Concluding Remarks

Collective religion as an overriding cultural and symbolic system, a kind of Croatian "holy canopy" that serves as an identification framework for the absolute majority of the population, also extends, to a certain extent, to nonreligious people. Religiosity as a framework of belonging undoubtedly affects the behavior, practices, and beliefs of nonreligious people and the ways in which they manage their "internal Other" status. In addition, collectivity can act as a push factor that drives some individuals out of the "religious canopy", either because of the coercion they oppose, the banalization that is present in such a form of religion on a broad social scale, or a "wrong" nationality or family origin.

Although collectivism in some cases can be a push factor, it is still highly valued in Croatian society, in which collective affiliations are very important (Jukić 1994; Vušković and Vrcan 1980). One way of being nonreligious in such a society is to keep it private and conform on the outside. In this way, although religion is intimately abandoned, it remains a social identity framework. Another form of keeping the connection with religion is "cherry-picking" elements of religiosity, which are then combined "in a manner that is perfectly consistent with the fluid subjectivities of modern society" (Turner 2010, p. 11). Although this individualized religiosity is a common occurrence in many other modern societies, in this case, it also serves as a link to collectivism. Emphasizing religious upbringing and former "insider status" can be used depending on the situation or passed on to children, by providing external protection under a "religious canopy". Individualization, thus, at least for some, arises within the symbolic framework of belonging, while transformation maintains a biographical continuity of identity.

Managing nonbelonging, as can be seen from the interviews, includes balancing between distancing oneself from socially and politically "wrong" religious attitudes and emphasizing one's own decision to leave religion, on the one side, and perceiving nonreligiosity as a certain social "handicap" or "stigma", on the other. Collective religion relies partly on exclusion, which interviewees try to avoid by adapting to the religious environment. At the same time, they feel "left out" and experience their nonreligiosity as lonely and isolating, which they often do not want to pass on to others. This creates a specific position for some nonreligious individuals, which is simultaneously "in" and "out" of religion, and challenges the way in which nonreligiosity is often imagined.

Although (non)religiosity is throughout this paper described more as an external form of belonging than as a substantively complex phenomenon, this does not mean that collective religion lacks a strong "spiritual" dimension. It also does not mean that religious collectivism is less important than the national one or that it is reduced, only to "serve" national identity. On the contrary, religiosity as a collective identity in its own right in the Croatian context is a phenomenon that has been "a source of primordial attachments for centuries" (Jakelić 2010, p. 28). Such a strong collective and public dimension to

religiosity makes the Croatian religious landscape necessarily a matter of (non)belonging. The observations made through this research reflect these long-lasting processes at an individual level. To what extent they are typical of the social contexts within collective religion can be seen only through more qualitatively-rich research of the (non)religious field in different social settings. On the other hand, the strong emotional charge that accompanies (non)religious and ethnic affiliation in Croatia still evokes traumatic historical and conflict experiences, specific to this context. It is, therefore, important to deconstruct essentialized (non)religious and ethnic stereotypes through research, and to address issues of collective identities, especially those related to religion, as areas that are still extremely sensitive in Croatian society.

**Funding:** This research received no external funding.

**Institutional Review Board Statement:** The study was approved by the Committee for the Evaluation of Research Ethics of the Department of Sociology of the Faculty of Humanities and Social Sciences Zagreb (Ethics Register Number 2016-38, approved on 28 December 2016).

**Informed Consent Statement:** Informed consent was obtained from all subjects involved in the study.

**Data Availability Statement:** The datasets obtained in this study are not open source. Requests may be made for access to the data presented in this study by contacting the author.

**Conflicts of Interest:** The author declares no conflict of interest.

## Notes

1.  Pew Research Center. Available online: https://www.pewresearch.org/fact-tank/2018/12/05/how-do-european-countries-differ-in-religious-commitment/ (accessed on 11 January 2022).

2.  Concepts of differentiated (Pickel et al. 2012) and contextual secularization (Pickel 2011; Pickel and Sammet 2012) focus on the idea that secularization's processes vary depending on the context and that they take place differently on different levels of society, at different times and with different intensities.

3.  Vatican Contracts include: 1. The Contract between the Holy See and the Republic of Croatia on the Spiritual Guidance of Catholics, Members of the Armed Forces and Police; 2. The Contract between the Holy See and the Republic of Croatia on Cooperation in the Field of Education and Culture; 3. The Contract between the Holy See and the Republic of Croatia on Legal Issues; 4. The Contract between the Holy See and the Republic of Croatia on Economic Issues. The relationship between the state and other religious communities was regulated only in 2002 when the government of the left-center passed the Law on the Legal Status of Religious Communities. According to this law, traditional religious communities had a simple registration process, while new religious communities had to meet certain prerequisites to be able to register (at least 500 members and registration as an association of citizens for at least 5 years before the registration as a religious community). These conditions contributed to the impression of the hierarchization of religious communities.

4.  Conservative civil initiative In the Name of the Family with the strong support of the Catholic Church initiated a referendum in 2013. The initiative was successful in its attempt to change the constitutional definition of marriage as a community between a man and a woman.

5.  Sermon of the monsignor Štambuk available online: https://www.hkv.hr/vijesti/hrvatska/5086-propovijed-mons-slobodana-tambuka-sa-sveanosti-proslave-dana-hrvatskih-muenika-na-udbini.html (accessed on 12 January 2022).

6.  (Tomislav Klauški 2019). Sekularizam je 'ušao u kuće'? Možda bježi pred klerikalizmom. 24 sata. Available online: https://www.24sata.hr/kolumne/sekularizam-je-usao-u-kuce-mozda-bjezi-ispred-klerikalizma-612293 (accessed on 21 January 2022); (Skoko 2013). Zašto u Hrvatskoj od samostalnosti nikada na čelu države nije bio uvjereni katolik? Večernji list. Available online: https://www.vecernji.hr/vijesti/zasto-u-hrvatskoj-od-samostalnosti-nikada-na-celu-drzave-nije-bio-uvjereni-katolik-912446 (accessed on 12 January 2022).

7.  Although the sample is biased in favor of the more educated, this doesn't present a significant limitation of the study. Namely, based on a qualitative methodology, this study tries to show the richness of individual nonreligious positions in a dominantly religious context but it doesn't strive for representativeness. Hence, the results presented here cannot be generalized to a larger population but can only try to illustrate different variations of individual experiences (not in an exhaustive way).

8.  Although the term "Croat-Catholic" is often generally, as well as in this quote, used in a mocking tone, it shows a lot of similarity with the Polish context and notion of „Polak-Katolik" as described in Trzebiatowska (2021).

9.  The term "cherry-picking" which is used throughout the text is quite similar to the much more recognizable concepts of "religion à la carte" or "religious bricolage" in the sense that they describe often idiosyncratic combinations of religious and nonreligious elements in individual beliefs and practices. However, those terms are "religiously oriented", while I wanted to accentuate

predominantly nonreligious orientations which still "borrow" religious elements. Hence, "cherry-picking" in this context would be more in line with the terms like "nonreligion à la carte" or "nonreligious bricolage".

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
