# Peer review of "Being Nonreligious in Croatia: Managing Belonging and Non-Belonging"

_religions, doi:10.3390/rel13050390_

Round 1
Reviewer 1 Report
Well done and well-structured paper. It contributes significantly to understanding non-religiosity and the construction of non- or partly- religious identity in a highly religious society. The paper corresponds well with the scholarship about the Croatian situation and the research on non-religiosity identities in various contexts. I recommend the paper to be published, with just a few minor suggestions.
The first one is to discuss what is called here “cherry-picking” elements of religiosity in relation to the well-known concept of “patchwork” or “religion à la carte”, often used in the sociology of religion to describe the complexity of the secularization and individualization process, and the way people manage their (non)religious identity.
The second one is to discuss results inside the findings that, though mainly collectivistic, the Croatian religious picture has important elements of secularization and individualization. While this is mentioned in the paper, it should be noted that many research pieces argue that there is a considerable level of “non-obedience” among believers, particularly in the field of moral issues. Does this, and how does this affect negotiating non-religious identity?
The third one is to discuss the possible limitation of the study, which comes from a particular sample consisting of persons with high educational backgrounds.
Author Response
Firstly, I would like to thank you for your time and effort to make this paper better.
Your suggestions were helpful and I made some revisions to the text according to them.
I will address them one by one.
1. The first one is to discuss what is called here “cherry-picking” elements of religiosity in relation to the well-known concept of “patchwork” or “religion à la carte”, often used in the sociology of religion to describe the complexity of the secularization and individualization process, and the way people manage their (non)religious identity.
Thank you for this comment. Surely, “religion à la carte”, or religious “bricolage” or “patchwork” are known terms in the sociology of religion. These terms are used in the context of religion for describing the process of individualization among the people who still consider themselves part of the “religious field”. I used different term (“cherry-picking”) because I thought that the above-mentioned terms are more “religiously oriented” and I wanted to show how people who define themselves as nonreligious still hold on to certain religious elements. I guess that terms like “nonreligion à la carte”, or nonreligious “bricolage”/ nonreligious “patchwork” could be good in this context. I tried to clarify this in note 9.
2. The second one is to discuss results inside the findings that, though mainly collectivistic, the Croatian religious picture has important elements of secularization and individualization. While this is mentioned in the paper, it should be noted that many research pieces argue that there is a considerable level of “non-obedience” among believers, particularly in the field of moral issues. Does this, and how does this affect negotiating non-religious identity?
Exactly, data show that there is a significant rejection of the Church's moral teachings among the believers in Croatia. Interviewees recognized this and addressed it as “empty Catholicism” or “quasi-Catholicism” (mentioned on page 7 and 8). This recognition in some situations was important “push factor” for some interviewees. For some of them, this notion makes it easier to conform and “fake religiosity”. I clarified this argument a bit more on page 8.
3. The third one is to discuss the possible limitation of the study, which comes from a particular sample consisting of persons with high educational backgrounds.
As this is a qualitative study that does not strive for representativeness but to show the richness of individual nonreligious positions in a dominantly religious context, I would argue that a biased sample in favor of the more educated, doesn't present a significant limitation of the study. In that sense, the results presented here cannot be generalized to a larger population (that is why I avoided using numbers or terms like “the majority of” and the like) but can only try to illustrate different variations of individual experiences (in a nonexhaustive way). On the other hand, various other Croatian and international studies showed that nonreligious people are in most cases more educated than average (Galen 2009; Marinović Jerolimov 1991; 1993; Hunsberger and Altemeyer 2006; Pasquale 2009; Zuckerman) and it is not unusual in qualitative research to gather a sample which includes more interviewees with higher education levels (Lee 2015; Frost 2012; Hazdovac Bajić 2019; Catto and Eccles 2013; Cimino and Smith 2007; 2011; Smith 2011; 2013; LeDrew 2013).
I added note 7 in which I addressed this question.
Reviewer 2 Report
This research tackles the issue of confessional self-identification of non-religious individuals in Croatian society who, on the other hand, are not members of any non-religious (e.g. atheist) organizations. In contrast to some organized forms of non-religiosity or atheism, the ‘quiet majority’ of the non-religious people with more diversified expressions of (non)-religiosity has stayed under radar in the sociological research. In Croatia, data gathered in the 2018 International Social Survey Program demonstrate, however, a much higher diversity among the non-religious respondents. The author argues that collective religion influences non-religiosity both by virtue of what is being rejected and what is selected to be preserved from it. Non-religious individuals sometimes experience their own attitude as lonely and isolating. This creates a specific position of some non-religious people who are simultaneously ‘in’ and ‘out’ of a religious confession and, thus, contests the way non-religiosity is portrayed in contemporary sociological studies.
The author’s proposals are grounded in the qualitative research and methodology (30 semi-structured interviews with snowball sampling). In Croatia, the locus of this sociological research, collective religion acts as a cultural and symbolic system that functions as an identity marker for absolute majority of the population. This also applies, at least to a certain extent, to non-religious persons. Accordingly, such a strong collective and public expression of religiosity makes the Croatian religious scenery a matter of (non-)belonging. The observations and insights made through this inquiry reflect these long-lasting processes on an individual level.
The author of this paper is well acquainted with religiosity in Croatia both as a scholar and as an “insider”. This contribution speaks across an interdisciplinary field of social sciences and humanities and has relevance for scholarship not only in sociology, but in social psychology and anthropology as well. A special quality of this paper is to be sought in its important interview data. Considering the paucity of articles in the domain of sociology of religion dealing with the profiles of non-religious individuals, this paper represents no small contribution to its own field of research. Therefore, I would recommend, without any reservation, that Religions publish this paper after minor interventions related to proofreading.
Author Response
I would like to thank you for your time and effort and positive feedback on the paper.
Reviewer 3 Report
An interesting topic. Expected results, but this confirmation of the expected also makes sense.
Author Response
I would like to thank you for your time and effort and overall positive feedback on the paper.
Reviewer 4 Report
This paper analyzes how nonreligious people in Croatia live their nonreligiosity within a context where religion is a source of collective belonging. Drawing on interview material, the author highlights some of the commonalities and differences across interviewees in the way they handle their (non-)belonging.
Overall, the paper is interesting and provides new insights into a context where being non-religious is a marker of difference. I think the paper is worth being published without much work.
I have two small suggestions. First, the section "Theoretical framework" doesn't really provide one. I would just call it "Conceptual background" or something along those lines. Second, I think the paper would gain some more strength if the author would make more explicit what is distinctively "Croatian" as opposed to the situation in other contexts. What makes this study relevant and what do we learn that is different from previous studies on nonreligious identities? In a way, the reference to the importance of the collective dimension of religion in the country already provides some insights. However, is this everything that makes the case special? One could think that this is not so different from other Catholic-majority countries where religion plays a similar role.
Author Response
Firstly, I would like to thank you for the time and effort you put into the review.
I found your suggestions helpful in making this paper better and I made some revisions to the text according to them.
I will address them individually below.
1. First, the section "Theoretical framework" doesn't really provide one. I would just call it "Conceptual background" or something along those lines.
I agree with this suggestion, “Conceptual Background” is a more suitable title for the chapter in question and I have changed it.
2. Second, I think the paper would gain some more strength if the author would make more explicit what is distinctively "Croatian" as opposed to the situation in other contexts. What makes this study relevant and what do we learn that is different from previous studies on nonreligious identities? In a way, the reference to the importance of the collective dimension of religion in the country already provides some insights. However, is this everything that makes the case special? One could think that this is not so different from other Catholic-majority countries where religion plays a similar role.
It is true that there are lots of similarities to other Catholic-majority countries, however, as you mentioned above, distinct collectivity and everything it brings is probably the biggest specificity. Unfortunately, there are not many other (qualitative) studies on this topic that would allow me to bring some conclusions about the specificities of the Croatian context. However, another thing that could be quite specific is a strong emotional charge that is connected with the issues of belonging to the “right” religion and “right” nation which reflects historical and war traumas. Religious and national collectivism in the process of disintegration of the former Yugoslavia were intertwined to accentuate the differences between conflicted parties while their emotional potential was used for political and war purposes. This dynamic between religious and national belonging that also implies a political standpoint exists as a stereotypical image in Croatian society. It is important to deconstruct such stereotypes through studies that show how religious/nonreligious, ethnic, and political divides are not that simple or straight-cut. I added a short paragraph at the end of the text to accentuate this.